# LDM²: A Large Decision Model Imitating Human Cognition with Dynamic Memory Enhancement

**Xingjin Wang[1,2], Linjing Li[1,2,3*], Daniel Dajun Zeng[1,2]**

[1]School of Artificial Intelligence, University of Chinese Academy of Sciences, Beijing, China
[2]State Key Laboratory of Multimodal Artificial Intelligence Systems,
Institute of Automation, Chinese Academy of Sciences, Beijing, China
[3]Beijing Wenge Technology Co.,Ltd
{wangxingjin2021, linjing.li, dajun.zeng}@ia.ac.cn

## Abstract

With the rapid development of large language models (LLMs), it is highly demanded that LLMs can be adopted to make decisions to enable the artificial general intelligence. Most approaches leverage manually crafted examples to prompt the LLMs to imitate the decision process of human. However, designing optimal prompts is difficult and the patterned prompts can hardly be generalized to more complex environments. In this paper, we propose a novel model named Large Decision Model with Memory (LDM²), which leverages a dynamic memory mechanism to construct dynamic prompts, guiding the LLMs in making proper decisions according to the faced state. LDM² consists of two stages: memory formation and memory refinement. In the former stage, human behaviors are decomposed into state-action tuples utilizing the powerful summarizing ability of LLMs. Then, these tuples are stored in the memory, whose indices are generated by the LLMs, to facilitate the retrieval of the most relevant subset of memorized tuples based on the current state. In the latter stage, our LDM² employs tree exploration to discover more suitable decision processes and enrich the memory by adding valuable state-action tuples. The dynamic circle of exploration and memory enhancement provides LDM² a better understanding of the global environment. Extensive experiments conducted in two interactive environments have shown that our LDM² outperforms the baselines in terms of both score and success rate, which demonstrates its effectiveness.

## 1 Introduction

The rapid development of large language models (LLMs) has led to remarkable revolution in the field of natural language processing (NLP). LLMs, such as Llama (Touvron et al., 2023), PaLM (Chowdhery et al., 2022), and GPT-4 (OpenAI, 2023), have achieved impressive results in a variety of tasks,

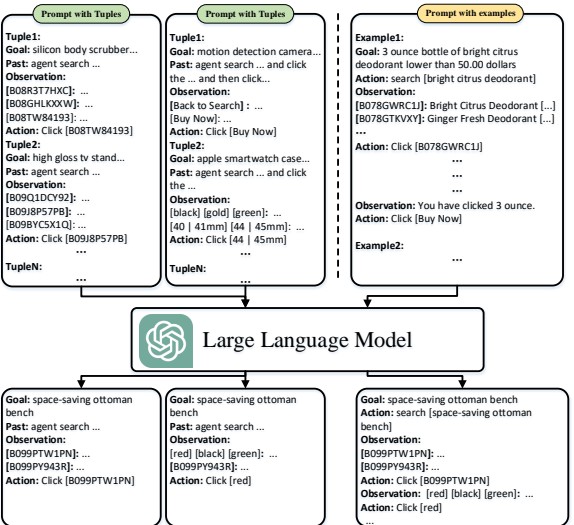

Figure 1: Comparison between prompt with examples and prompt with state-action tuples.

including text/code generation (Chen et al., 2021), question answering (Zhou et al., 2022), and reasoning (Wei et al., 2022b), to name a few. Besides NLP tasks, LLMs are also employed as policy agents to accomplish decision-making tasks (Kim et al., 2022; Mialon et al., 2023). Nowadays, most approaches adopt the standard prompting paradigm which uses manually crafted in-context examples to prompt the LLMs (Sanh et al., 2022; Wei et al., 2022a) making decisions. However, the standard prompting is not suitable for decision-making tasks, as it restricts the LLMs to merely imitate the provided examples, making the generated decisions highly context-sensitive. Worse still, standard prompting cannot generate admissible decisions in complex environments (Liu et al., 2021; Dong et al., 2022). In certain cases, the LLMs may fall into confusion when new situations are greatly different from the examples. In more challenging situations, even humans are unable to provide complete solution examples. Meanwhile, once the prompt examples are fixed, the LLMs can no longer

---

*Corresponding author

learn from the feedback of the environment, thus cannot further improve the performance.

In this paper, we proposed **L**arge **D**ecision **M**odel with **M**emory (LDM$^2$), a framework that enhances the standard LLMs with dynamic updating memory. The memory mechanism maintains the most valuable state-action tuples when imitating human decisions. As shown in Fig. 1, the prompts generated by the proposed LDM$^2$ focus on providing LLMs with sufficient information to guide decision making in the current situation, rather than relying solely on the simple examples in the entire process of decisions. In order to obtain sufficient state-action tuples in the memory, LDM$^2$ incorporates a memory formation stage that is analogous to the traditional imitation learning (Hussein et al., 2017). In this stage, we take human trajectories as training data and instruct the LLMs to produce numerous standard state-action tuples including the task goals, observations, historical information, and actions. These standard state-action tuples are preserved to form the initial memory. In the inference phrase, we retrieve the most similar state-action tuples from the memory with current observation and then construct the prompt to inspire the LLMs dynamically. These retrieved state-action tuples 1) inform the LLMs which actions would be taken by human in the current state and 2) help the LLMs understand the global environment.

Differentiate from traditional imitation learning (Hussein et al., 2017), LDM$^2$ is equipped with a dynamic memory refinement stage to enhance the memory with valuable state-action tuples. First, We conduct tree exploration to generate all potential decision processes and evaluate them according to the environment rewards. Then, we add the state-action tuple corresponding to the best decision process into the memory. This exploration-evaluation-adding circle mimics the traditional reinforcement learning framework (Arulkumaran et al., 2017; Yao et al., 2020). The refinement stage not only expands the action space of the LLMs, but also enable the LLMs to deal with new situations not covered by the initial memory.

We evaluate the proposed LDM$^2$ in two interactive environments: WebShop and ALFworld. LDM$^2$ outperforms the standard few shots prompting methods and other methods prompted with verbal reasoning. We further analyze the successful examples in both tasks and find that LDM$^2$ has a more diverse action space compared with methods using fixed examples prompt, this advantage empowers the LLMs to handle unseen or complex situations. Additionally, We conduct ablation experiments to evaluate the memory refinement mechanism. The results demonstrate the effectiveness of adding highly rewarded state-action tuples into the memory. Our main contributions can be summarized as:

- We propose a novel paradigm that leverages a two-stage memory mechanism to dynamically prompt the standard LLMs for decision-making tasks.

- We make full use of the standard LLMs in the memory formation stage to produce state-action tuples and generate the corresponding indices.

- We adopt tree exploration to generate potential decision processes and instruct the LLMs to identify the most valuable state-action tuples to enhance the memory.

## 2 Related work

Our paper is closely related to the following three research directions: LLMs for decision-making, feedback for LLMs, and memory and retrieval for LLMs. In this section, we briefly review the literature on these research.

**LLMs for decision-making** Powerful LLMs are able to act as policy models to make decisions in interactive environments. Li et al. (2022) constructed a general framework for decision-making that uses LLMs to encode observations, goals, and history and then generate actions. Demonstration of example prompts and utilization of high-level function libraries are employed to explore innovative strategies (Huang et al., 2022a; Liang et al., 2022; Wu et al., 2023; Nakano et al., 2021; Shen et al., 2023). Prompting structure with pre-defined functions, behaviors, and examples are leveraged to ground LLMs to generate robotic actions (Ahn et al., 2022; Huang et al., 2022b; Vemprala et al., 2023). However, these methods use manually crafted examples to prompt the LLMs which results in decision-making in a fixed direction. Our LDM$^2$ leverages dynamic state-action tuples as prompt to improve the effectiveness of decisions.

**Feedback for LLMs** Recent techniques have emerged that focus on establishing closed-loop systems which are capable of utilizing the scalar or

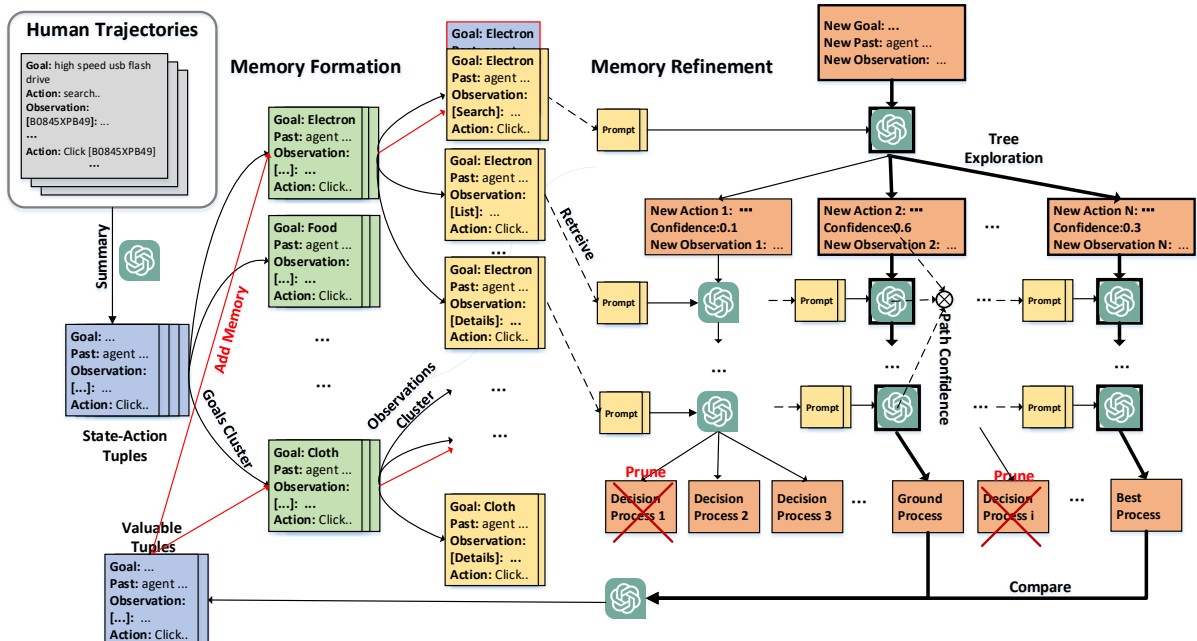

Figure 2: Overview of the proposed LDM$^2$. The formation of memory (left) takes human trajectories as training data to produce standard state-action tuples (blue blocks) and then instructs the LLMs to generate the index by clustering data based on the goals (green blocks) and observations (yellow blocks). The refinement of memory (right) leverages the tree exploration to produce potential action process and instruct the LLMs to find the most valuable state-action tuples. Finally, the valuable tuples are added into (red arrow) the corresponding memory batch.

textual feedback from environment or human to update the LLMs (Christiano et al., 2017; Ouyang et al., 2022; Chen et al., 2022; Bai et al., 2022). Madaan et al. (2023) and Pryzant et al. (2023) employ an iterative framework for self-refinement to optimize the prompt of LLMs based on the feedback of self-evaluation. REFINER (Paul et al., 2023) fine-tunes another critic model to provide intermediate feedback within trajectories to improve the reasoning response. ReAct (Yao et al., 2022b) prompts LLMs with both verbal reasoning traces and actions which guides the models to perform dynamic reasoning according to environmental feedback. Reflexion (Shinn et al., 2023) converts binary or scalar feedback from the environment into verbal feedback which is then added in the prompt of the next episode. Introspective Tips (Chen et al., 2023) learns tips from the action trajectories and environmental feedback to empower the LLM agents with self-optimizing capabilities. These approaches mainly aim to leverage reward feedback to augment the prompt of LLMs, however, LDM$^2$ adds highly rewarded state-action tuples into the memory which achieves dynamic learning ability.

**Memory for LLMs** Memory could store information perceived from the environment and leverages the recorded memories to facilitate future actions. Generative Agents (Park et al., 2023) maintain a memory stream to record the experience including observations and behaviors. Reflexion (Shinn et al., 2023) stores experiential feedback in natural language within a sliding window. Voyager (Wang et al., 2023) employs natural language descriptions to represent skills within the Minecraft game, which are directly stored in memory. MemoryBank (Zhong et al., 2023) encodes the memory segment into embedding vector which could enhance memory retrieval and reading efficiency. Knowledge base is also used as the memory to retrieve relevant information and construct the task-related augmented prompts (He et al., 2022; Trivedi et al., 2022; Khattab et al., 2022). Our LDM$^2$ also constructs a memory to record vast state-action tuples, which forms the retrieval indices list through instructing the LLMs to cluster different goals and observations.

## 3 Methodology

In this paper, we consider a general setup of LLMs as policy models to accomplish decision-making tasks in an interactive environment. In the follow-

ing, $LLM_A(\cdot)$ denotes employing a LLM to perform the $A$ function/operation.

## 3.1 Problem Definition

We leverage $N$ human decision trajectories $\mathcal{T} = \{t^1, t^2, \cdots, t^N\}$ as the training data, where each trajectory $t^i = \{o_1^i, a_1^i, \cdots, o_{T_i}^i, a_{T_i}^i\}$ has a task goal $g^i$, $T_i$ is the length of this trajectory, $o_\tau^i$ is the observation at time step $\tau$, $a_\tau^i$ is the human action when faced with $o_\tau^i$, $1 \le \tau \le T_i$. The set $\mathcal{T}$ of trajectories can be further processed into a memory $\mathcal{M}$ consisting of standard state-action tuples:

$$\mathcal{M} = \left\{ \left\langle g^i, h_\tau^i, o_\tau^i \to a_\tau^i \right\rangle | 1 \le i \le N, 1 \le \tau \le T_i \right\},\tag{1}$$

where $h_\tau^i$ (to be further elaborated in the next subsection) represents historical information about the observations and actions before time step $\tau$. The memory $\mathcal{M}$ provides the LLMs with a sufficient set of state-action tuples to help generate proper actions in various situations. Based on $\mathcal{M}$, the LLMs can interactively explore the environment. Given a new task goal $g^j$, the LLMs receive the current observation $o_\tau^j$ at time step $\tau$, the historical information $h_\tau^j$ and the context prompt $p_\tau^j$ to generate an action $a_\tau^j$:

$$a_\tau^j = LLM(g^j, h_\tau^j, o_\tau^j, p_\tau^j),\tag{2}$$

where the context prompt $p_\tau^j$ is a subset of memory $\mathcal{M}$ that can be retrieved from $\mathcal{M}$ according to the task goal $g^j$ and the current observation $o_\tau^j$:

$$p_\tau^j = LLM_{retrieve}(\mathcal{M}|g^j, o_\tau^j).\tag{3}$$

In order to accomplish the goal $g^j$, the context prompt tells the LLMs about actions taken by human in the current state, invoking the LLMs to comprehend the environment. Subsequently, the LLMs generate the complete decision process $P^j$ and get the final reward $r$:

$$P^j = \left\{ g^j; o_1^j, a_1^j, \cdots, o_T^j, a_T^j \right\}.\tag{4}$$

Due to the context length, an intrinsic limitation of LLMs, we have to partition all those $N$ trajectories into $n$ batch data with size $B$, where $N = nB$. Each batch data $\mathcal{T}_b = \left\{ t^{(b-1)B+1}, t^{(b-1)B+2}, \cdots, t^{Bb} \right\}, b = 1, 2, \cdots, n$, is processed to form a batch memory $\mathcal{M}_b$ by the same procedure described above. Accordingly, we construct $n$ independent batch memory $\mathcal{M}_1, \mathcal{M}_2, \cdots, \mathcal{M}_n$. Each batch memory $\mathcal{M}_b$

assists the LLMs completing a whole decision process $P_b^j$ w.r.t. the goal $g^j$. We require the LLMs to choose the optimal process as the final decision:

$$P_{final}^j = LLM_{choose} \left( P_1^j, P_2^j, \cdots, P_n^j \right),\tag{5}$$

where $P_b^j$ is the decision process based on the batch memory $M_b$, $b = 1, 2, \cdots, n$.

## 3.2 Memory Formation

Previous imitation learning methods enable a policy agent to mimic expert behavior through updating the parameters of language models. However, in the new prompt paradigm of LLMs (Liu et al., 2023), we need to integrate human cognition into the context prompt while freezing the parameters of LLMs. Our LDM$^2$ leverages the memory to store vast state-action tuples and constructs dynamic context prompt based on the current observation, which imitates the human decisions.

**Memory Structure and Format** As introduced in the last subsection, the memory $\mathcal{M}$ consists of a large number of standard state-action tuples. This subsection depicts how to construct the memory $\mathcal{M}$. Given a human trajectory $t^i = \left\{ o_1^i, a_1^i, \cdots, o_{T_i}^i, a_{T_i}^i \right\}$, it can be divided into $T_i$ standard state-action tuples. In addition to the current observation $o_\tau^i$, the past decision process $\{o_1^i, a_1^i, \cdots, o_{\tau-1}^i, a_{\tau-1}^i\}$ and the task goal $g^i$ are the crucial factors that LLMs must consider when making decision at time step $\tau$. However, the raw data of the past process is relatively long as the prompt context. Therefore, we instruct the LLMs to summarize them into the brief historical information:

$$h_\tau^i = LLM_{summary}(o_1^i, a_1^i, \cdots, o_{\tau-1}^i, a_{\tau-1}^i; g^i).\tag{6}$$

The instruction of this summary process require the LLMs to briefly describe the past experiences and assess the progress of tasks in the current state. The historical information also provide LLMs agent with planning information, indicating the decision stage it has reached and assisting the agent in making proper decisions. The complete prompt of the summary process is listed in the appendix A.

To sum up, as indicted in Eq. (1), a standard state-action tuple of human trajectory $t^i$ at time step $\tau$ contains four elements: task goal $g^i$, agent history $h_\tau^i$, current observation $o_\tau^i$, and the current action $a_\tau^j$ demonstrated in Eq. (2). As shown in Fig. 2,

the blue blocks in the left represents the obtained state-action tuples from the human trajectories.

**Memory Index**   To efficiently store and retrieve tuples from the memory $\mathcal{M}$, we construct the following index system including two types of indices: goal index and observation index.

First, we cluster the goals of different tasks in each batch memory to form its goal index, which is achieved by instructing the LLMs to generate high-level types of received information and classify each task goal to the corresponding type:

$$index_g^b = LLM_{cluster}\left(g_b^1, g_b^2, \cdots, g_b^B\right), \quad (7)$$

where $index_g^b$ is the goal index of batch memory $M_b$, $b = 1, 2, \cdots, n$ and $g_b^\ell$ is the goal of trajectory $t^{(b-1)B+\ell}$, $\ell = 1, 2, \cdots, B$. Then, we cluster the observations of each goal type by instructing the LLMs to classify all observations into a high-level type to form the observation index:

$$index_o^{bk} = LLM_{cluster}\left(o_1^{bk}, \cdots, o_{Z_{bk}}^{bk}\right), \quad (8)$$

where $index_o^{bk}$ is the observation index of goal type $k$ in the batch memory $M_b$ and $Z_{bk}$ denotes the total quantity of observations in goal type $k$.

In the inference phrase, the LLMs agent firstly leverage the goal index to retrieve the similar tasks and then use the observation index to find the similar situations with the current state. The complete prompt of the cluster process is listed in appendix A. As shown in Fig. 2, the green and yellow blocks are the classified data based on the goals and observations. Compared with traditional clustering methods, LLMs based clustering could work with text-based input and generate text-based output instead of numerical data representations, which is more flexible and effective to capture complex semantic relationships in text-rich decision environment.

### 3.3   Memory Refinement

The above memory formation stage follows the imitation learning paradigm, which provides LDM$^2$ a initial policy. To improve the policy dynamically, we adopt tree exploration, which mimics online reinforcement learning, to enhance the memory by adding the most valuable state-action tuples into $\mathcal{M}$.

**Tree Exploration**   We leverage the tree exploration to generate more possible decision processes

through splitting more leaf nodes at each parent nodes. For task goal $g^i$, at each time step $\tau$, we instruct the LLMs to provide some possible actions based on the current observation and the memory $\mathcal{M}$, and we prompt the LLMs to assign a confidence score to each action (the complete prompt is listed in appendix A).

$$a_\tau^{j_1}, c_\tau^{j_1}, ..., a_\tau^{j_z}, c_\tau^{j_z} = LLM(g^j, h_\tau^j, o_\tau^j, p_\tau^j), \quad (9)$$

If the action distribution of retrieved state-action tuples in each node is highly concentrated, the LLMs will select the majority action and proceed to the next state. Otherwise, the LLMs will retain all admissible actions and explore a subtree for each action. Meanwhile, we maintain the confidence of each exploration path, which is the product of confidence score of all nodes along the path. To avoid the exponential growth of exploration paths, we only retain top-n (n=4) confidence paths for the the next step of exploration and prune the valueless exploration paths. Finally, we obtain top-n (n=4) decision processes and get the final rewards at the leaf nodes.

**Memory Enhancement**   The tree exploration generates a set of decision process $P_1^i, P_2^i, \cdots$ with reward $r_1, r_2, \cdots$, respectively. The process with the maximal reward is the best decision process $P_\star^i$, and the process which is formed by instructing the LLMs to generate one best decision process based on the memory $\mathcal{M}$ is the ground decision process $P_g^i$. If $P_\star^i$ have a higher reward than $P_g^i$, we then instruct the LLMs to compare these processes and find the key decision step in the $P_\star^i$. Then we treat the subtrajectory after the key steps as the valuable data to enhance the memory:

$$\{o_{\tau^\star}^i, a_{\tau^\star}^i, \cdots, o_{T_i}^i, a_{T_i}^i\} = LLM_{compare}(P_\star^i, P_g^i), \quad (10)$$

where $\tau^\star$ is the key step in the best decision process given by the LLMs.

The obtained valuable subtrajectory is converted into standard state-action tuples using the method depicted in the above sections. For each pair, we leverage the goal index and observation index to find the corresponding categories and directly add this tuples into this subset of the memory. Adding these new valuable state-action tuples into the memory $\mathcal{M}$ changes the distribution of the action space. As shown in the right of Fig. 2, the LLMs conduct tree exploration to generate more decision processes. The best and ground processes in the

leaf nodes are converted to state-action tuples to enhance the memory $\mathcal{M}$ which are marked by the red arrows.

## 4 Experiment Setup

### 4.1 Tasks and Datasets

We evaluate $LDM^2$ on two language-based interactive decision-making tasks: ALFWorld and WebShop. Both are complex environments with various observations and actions that are difficult to be addressed through fixed examples prompt.

**WebShop** WebShop (Yao et al., 2022a) is a simulated e-commerce website environment with real-world products and crowd-sourced text instructions. Given a text instruction specifying a product requirement, an agent needs to navigate multiple types of webpages and make actions to find, customize, and purchase the required product. The performance is evaluated by average score and sucess rate, the former the percentage of desired attributes covered by the chosen product averaged across all episodes, and the latter is the percentage of episodes where the chosen product satisfies all requirements on 500 test instructions.

**ALFworld** ALFworld (Shridhar et al., 2020) is a suite of text-based environments which require the agent to accomplish multi-step tasks in a variety of interactive environments based on TextWorld (Côté et al., 2019). ALFworld includes six types of goals (e.g. picking specified objects and putting in designated place, examining objects by specific instructions and manipulating the objects through specific means). The agent needs to navigate and interact with a simulated household to determine the actions. We conduct experiments on 134 test games, the result is scored by the success rate that is the percentage of episodes which achieves the given goals.

### 4.2 Baselines

We compare $LDM^2$ with two prompt-based approaches using complete examples and traditional imitation learning methods trained with annotated trajectories. In all experiments, we employ GPT-3.5 (`gpt-3.5-turbo`) as the workhorse LLM.

**Standard** We directly leverage few-shot successful example decision processes as context to prompt the LLMs for both tasks.

| Method | Score | SR |
|---|---|---|
| Standard | 62.7 | 29.6 |
| WebShop | 62.4 | 28.7 |
| ReAct | 67.1 | 39.6 |
| Reflexion | 68.4 | 40.2 |
| $LDM^2_{In}$ | 71.8 | 40.8 |
| $LDM^2_{In+Re}$ | **72.4** | **41.6** |
| Human Expert | 82.1 | 59.6 |

Table 1: Score and success rate results on WebShop. The results of WenShop are from (Yao et al., 2022a). The trial number of Reflexion is 4. The $LDM^2_{In}$ is the result of initial memory and $LDM^2_{In+Re}$ is the result of refined memory.

**ReAct** The prompts in the ReAct (Yao et al., 2022b) include not only the observations and actions, but also verbal reasoning traces to guide the LLMs perform dynamic reasoning in the decision process.

**Reflexion** Reflexion (Shinn et al., 2023) is an extension of ReAct, which uses self-reflect to convert binary or scalar feedback from the environment into verbal feedback and repeats the same task many times based on the reflection.

**Imitation Learning** BUTLER (Shridhar et al., 2020) is an imitation learning agent for ALFWorld tasks trained on a large amount of human trajectories. WebShop (Yao et al., 2022a) finetunes multiple language models to learn how to search and choose from various shopping processes.

### 4.3 Training Setup

In the $LDM^2$ memory formation stage, we use 500 human shopping trajectories to construct the WebShop training memory and 200 expert trajectories for each task in the ALFWorld to form the memory. We set the batch size as 100 for WebShop and 50 for ALFWorld to construct multiple independent batch memories for both tasks. In the $LDM^2$ memory refinement stage, we utilize 100 new instructions to explore the WebShop environment and 10 new goals for each task to explore the ALFWorld environment.

## 5 Results and Analyses

### 5.1 WebShop

As shown by Tab. 1, our $LDM^2$ outperforms all baselines in both score and success rate, which indicates the effectiveness of leveraging state-action

| Method | Pick | Clean | Heat | Cool | Look | Pick2 | All |
|---|---|---|---|---|---|---|---|
| Standard | 88 | 55 | 70 | 67 | 72 | 41 | 66 |
| $\text{BUTLER}_{\text{g}}$ | 33 | 26 | 70 | 76 | 17 | 12 | 22 |
| BUTLER | 46 | 39 | 74 | **100** | 22 | 24 | 37 |
| ReAct | 63 | 48 | 74 | 71 | 67 | 35 | 60 |
| $\text{ReAct}_{\text{best}}$ | 92 | 65 | **96** | 86 | 78 | 47 | 78 |
| Reflexion | 88 | 81 | 83 | 90 | 83 | **88** | 85 |
| $\text{LDM}^2_{\text{In}}$ | 88 | 81 | 87 | 90 | 83 | 71 | 84 |
| $\text{LDM}^2_{\text{In+Re}}$ | **96** | **87** | 91 | 90 | **89** | 76 | **89** |

Table 2: ALFWorld task-specific success rates(%). BUTLER and $\text{BUTLER}_{\text{g}}$ results are from (Shridhar et al., 2020). ReAct use two examples as prompt and $\text{ReAct}_{\text{best}}$ is the best result in 6 prompts(Yao et al., 2022b). The trial number of Reflexion is 5. The $\text{LDM}^2_{\text{In}}$ is the result of initial memory and $\text{LDM}^2_{\text{In+Re}}$ is the result of refined memory.

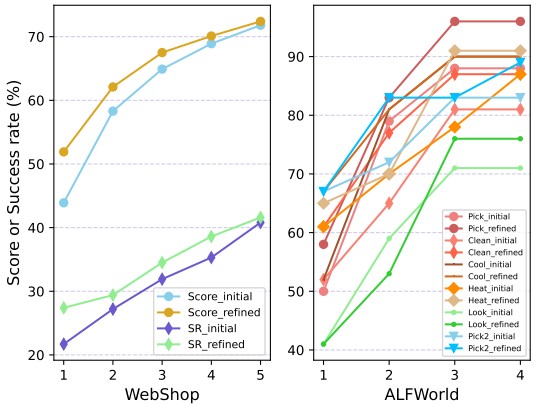

Figure 3: The relationship between number of batch data and score or success rate.

tuples as dynamic prompts to instruct LLMs to make decisions. The traditional Imitation Learning method finetunes a language model with medium size, which results in poorer performance compared with the prompt-based method using LLMs. Also, the poor performance of the standard prompting method validates that fixed prompt of complete examples is not suitable for decision-making tasks. Standard prompting methods may select products in the same order as demonstrated in the examples, instead of selecting the proper products according to the instructions. Additionally, they may generate actions that similar to the examples but not executable in the current environment. Though ReAct adds dynamic reasoning into the prompt, it may make incorrect reasoning when encountered with different situations. For instance, the ReAct prompts include thought process of clicking corresponding attribute options, thus it may imitate the given reasoning process to click one option even when no option exists in the current situation.

**Analysis on the Initial Memory** The result of experiment on initial memory outperforms other baselines, which demonstrates the effectiveness of using initial memory to provide sufficient state-action tuples for the LLMs. State-action tuple examples of the current situation assist the LLMs in understanding the patterns of the environment instead of merely imitating the actions of existing examples. As shown in the upper left of Tab. 3, ReAct first selects one of the most plausible product from the list of products, but the product detail is not fully aligned with the instruction. However, the LLMs still imitate the examples to purchase this one. The proposed $\text{LDM}^2$ also select this product, but it clicks the "Prev" button to search another one after finding that the detail is not matched. The reason is that the retrieved memory in this state includes same situations in which the products are not the best choice, and a click on "Prev" to find another product is illustrated in the examples.

**Analysis of the Refined Memory** The performance of model equipped with the refined memory is better than the one with only the initial memory, which shows the effectiveness of updating the memory by exploring more valuable decision processes. The nodes which generate multiple actions in the tree exploration are mainly the products selection nodes. As shown in the upper right of Tab. 3, based on the initial memory, $\text{LDM}^2$ selects the most plausible products, but these products do not include the specified options (size 11 women), which results in a lower reward. However, $\text{LDM}^2$ with the refined memory explores more possible products and finds the product matching all options. In addition, the tree exploration can expand and enrich

| | | |
|---|---|---|
| | **Instruction:** *I am looking for a high power sound column subwoofer, that uses bluetooth and is also a 3d surround sound system, and price lower than 650.00 dollars.* | **Instruction:** *I need khaki steel toe shoes in size 11 women, and price lower than 70.00 dollars.* |
| **WebShop** | **ReAct:** Search [high power subwoofer···]→Click [···RRQ] (··· Bluetooth Speaker subwoofer Portable Small 3D Surround···)→Click [Buy Now]→Reward [0.5] | **Initial:** Search [khaki steel toe shoes size 11 women]→Click [···LJH]→Click [khaki]→Click [10.5 women \| 9 men] (Size ···[10.5 women \| 9 men] [11.5 women \| 9.5 men]···)→Click [Buy Now]→Reward [0.75] |
| | **Initial:** Search [high power sound column subwoofer ···]→Click [···RRQ]→Click [< Prev]→Click [···TS6] (··· Bluetooth Speaker 60W High Power Sound Column Outdoor Subwoofer 3D Stereo Surround Sound System )→Click [Buy Now]→Reward [1.0] | **Refined:** Search [khaki steel toe shoes size 11 women]→Click [···Q2K]→Click [khaki]→Click [11 women \| 9 men] (Size ···[10.5 women \| 8.5 men] [11 women \| 9 men]···)→Click [Buy Now]→Reward [1.0] |
| **ALFWorld** | **Goal:** *You are in the middle of a room, put two soapbar in garbagecan.* | **Goal:** *You are in the middle of a room, cool some mug and put it in cabinet.* |
| | **ReAct:** go to cabinet 1→open cabinet 1→go to cabinet 2→go to cabinet 3→go to cabinet 3→go to cabinet 4→open cabinet 4→take soapbar 2 from cabinet 4→···put soapbar 2 in/on garbagecan 1→···go to cabinet 1→···Reward[0] | **Initial:** go to fridge 1→go to shelf 1→go to cabinet 1→go to cabinet 2→go to cabinet 3→go to cabinet 4→go to cabinet 5→go to cabinet 6→··· ReWard[0] |
| | **Initial:** go to cabinet 1→open cabinet 1→go to countertop 1→take soapbar 1 from countertop 1→···put soapbar 1 in/on garbagecan 1→go to countertop 1→take soapbar 3 from countertop 1→···put soapbar 3 in/on garbagecan 1→Reward[1] | **Refined:** go to fridge 1→go to countertop 1→take mug 2 from countertop 1→go to fridge 1→cool mug 2 with fridge 1→go to cabinet 1→put mug 2 in/on cabinet 1→Reward[1] |

Table 3: Sample result of WebShop and ALFWorld based on the initial memory and refined memory.

the initial memory to help the model revise actions in case the state-action tuples in the initial memory are not sufficient.

## 5.2 ALFWorld

According to Tab. 2, LDM$^2$ also outperforms all baselines evaluated on tasks in ALFWorld. The prompt-based LLMs outperform the traditional deep learning method. In ALFWorld, finding the desired object and performing correct operations often involves many steps, which results in a loss of the current state's tracking when received prompt with long examples. Despite its dynamic reasoning ability, ReAct still generates incorrect actions due to the long decision process and unseen situations.

**Analysis on the Initial Memory** The imitation learning has a better performance than other baselines, which shows LDM$^2$ can form a valid initial memory for ALFWorld tasks. In the memory formation stage, the LLMs not only cluster the household items into many high-level types like furniture, kitchen ware, and electronic devices, but also cluster the observations as kitchen room, bathroom, and bedroom. The state-action tuples in different subsets of the memory guide LDM$^2$ to go to the most likely place to find the desired object, take the desired object, manipulate the object correctly, and then put the object in the designated place. As

shown in the bottom left of Tab. 3, ReAct takes first sopabar from cabinet and then falls into confusion, as it does not know where to find the second item. Based on the memory, the human experience teaches LDM$^2$ to go to the most likely place (countertop) instead of exploring all places, thus finds the items efficiently.

**Analysis on the Refined Memory** The nodes split in the ALFWorld are the most possible selection nodes. In the memory refinement stage, LDM$^2$ generates some possible places to explore the environment. LDM$^2$ finds more convenient and fast ways to complete the goals and adds these tuples into the memory. Meanwhile, in some cases where there are no analogous situations to the test task in the memory, the tree exploration process can assist the LLMs in exploring the common appearing places of the unseen items. As shown in the bottom right of Tab. 3, LDM$^2$ fails to find the desired item in the new environment based on the past human experience, but the tree exploration help find the item by providing more possible places.

## 5.3 Analysis of Training Data

As the batch data represents human experience, we expect that the performance of LDM$^2$ will increase as the data size increases. Thus we conduct experiment to find the relationship between score/success

rate and the size of the batch data.

As shown by Fig. 3, the score/success rate increases in both tasks as the size of batch data increases, which demonstrates that more human trajectories can enhance the LLMs' ability to make more proper actions in the current state. In WebShop, more data means more types of products to help the LLMs learn what to search or click. In ALFWorld, more data provides the LLMs more information about where the desired objects may appear and how to manipulate them correctly.

## 6   Conclusion

This paper proposed $LDM^2$, which enhances the standard LLMs with dynamic updating memory to maintain the most valuable state-action tuples to imitate human decision. $LDM^2$ consists of two stage: memory formation and memory refinement. In the formation stage, we take human trajectories as training data and instruct the LLMs to produce numerous standard state-action tuples. These standard state-action tuples are preserved to form the initial memory. $LDM^2$ is equipped with dynamic memory refinement stage to enhance the memory through adding valuable state-action tuples. We conduct tree exploration to generate all potential decision processes and add the state-action tuple corresponding to the higher reward decision process into the memory. Experiments on two interactive environments illustrated that $LDM^2$ outperforms the standard few shots prompting methods and the ablation study verified the effectiveness of the memory formation and refinement mechanism.

## Limitations

To fully leverage the capabilities of $LDM^2$, we need to collect a certain amount of high-quality human trajectories, which may be difficult and infeasible in some environments. Hence, we need to stimulate the LLMs' own reasoning and understanding ability when having few data to interact with the environments.

Meanwhile, $LDM^2$ bears a higher time cost compared with other standard prompting methods. In the inference phrase, $LDM^2$ needs to retrieve relevant state-action tuples in the memory, which results in a significant lower action generation speed. However, the computational cost of our method is comparable with other methods, because the prompt length of each timestep of our method is shorter and only the current state needs to be con-

sidered, but other methods must record the whole past experience which will increase as time goes by. The amount of inference tokens of our method and existing methods are roughly equal.

## Acknowledgements

This work was supported in part by the Strategic Priority Research Program of Chinese Academy of Sciences under Grant #XDA27030100 and the National Natural Science Foundation of China under Grants #72293573, #72293575 and #62206282.

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

# A   Prompt Details

All the prompts used in this paper to instruct the LLMs are shown in Tab. 4.

# B   Example Process

Tab. 5 and Tab. 6 provide an example decision process in the WebShop task produced by the proposed LDM$^2$. Due to the long decision-making process, we omit some parts of prompts.

| | |
|---|---|
| **Evlaution:** | *I will give you a task goal and agent past action process.* |
| | *You should partition the goal into some subgoal and judge the past actions whether complete these subgoals.* |
| | *The desired format is:* |
| | *subgoal 1:goal - complete or in complete* |
| | *etc.* |
| | *Do not give me explanation.* |
| **Summarization:** | *I will give you the past process and you should summarize the past process.* |
| | *The desired format must be:* |
| | *Summary:* |
| | *Do not give me explanation.* |
| **Cluster Goals:** | *I will give you a few numbered task goals.* |
| | *You need to help me classify these goals into some types. The number of each category should be almost average. The category must be high-level type.* |
| | *The desired format is:* |
| | *High-level Type1: type name [number]* |
| | *High-level Type2: type name [number]* |
| | *High-level Type3: type name [number]* |
| | *etc.* |
| **Cluster Observations:** | *I will give you a few numbered observations.* |
| | *You need to help me classify these observations into some types. The number of each category should be almost average. The category must be high-level type.* |
| | *The desired format is:* |
| | *High-level Type1: type name [number]* |
| | *High-level Type2: type name [number]* |
| | *High-level Type3: type name [number]* |
| | *etc.* |
| **Index Observations:** | *I will give you some numbered goal types and examples of this type.* |
| | *You should judge the new goal belongs to which type.* |
| | *The desired format is:* |
| | *[Type number]: reason* |
| | *Do not give me other information.* |
| **Index Observations:** | *I will give you some observation numbered types and examples of this type.* |
| | *You should judge the new observation belongs to which type.* |
| | *The desired format is:* |
| | *[Type number]: reason* |
| | *Do not give me other information.* |
| **Action:** | *I give some numbered examples and a new observation.* |
| | *You should imitate the actions in the example and give me the next action.* |
| **Tree Exploration:** | *I give some numbered examples and a new observation.* |
| | *You should imitate the actions in the example and give me some possible next actions and the confidence of each action. All confidence should sum equal 1.* |
| **Compare:** | *I will give you two shopping process. Each process have some numbered step.* |
| | *The first process is better than the second, Can you tell me the first number that two process different.* |
| | *The desired format is:* |
| | *Number: Reason.* |
| | *Do not give any other information and strictly follow the format.* |

Table 4: Prompt used in the LDM$^2$.

**Prompt1:**

1. Instruction: i need a pair of shoes with rubber soles. remember to get size seven and a half womens, and price lower than 40.00 dollars.

Past:

Action: search[shoes, rubber soles, size 7.5, women's less than 40]

2. Instruction: can you find some carol wright mens lounge pants in a 5xl? i want the ones with a draw string closure that are charcoal colored, and price lower than 40.00 dollars

Past:

Action: search[5xl carol wright men's lounge pants charcoal color]

3. Instruction: i need some black ankle strap flats that are in a size 9 wide, and price lower than 60.00 dollars

Past:

Action: search[black ankle strap flats in a "size 9 wide"]

4. Instruction: i need a fleece jacket for the winter that is warm and gray, and price lower than 40.00 dollars

Past:

Action: search[winter fleece jacket gray warm]

5. Instruction: i want a extra large yellow mens loose fit shirt, and price lower than 40.00 dollars

Past:

Action: search[extra large yellow men's loose fit shirt]

**Observation1:** Instruction: *Find me long lasting, moisture wicking, loose fit men's shorts with elastic waistband, quality materials, polyester cotton, short sleeve with color: black, and size: 4x-large, and price lower than 50.00 dollars.*

**Action1:** search[long lasting moisture wicking loose fit men's shorts with elastic waistband black 4x-large polyester cotton]

**Prompt2:**

1. Past: Searched for 5XL Carol Wright men's lounge pants in charcoal color.

The interface is:

[Back to Search]

Page 1 (Total results: 50)

[Next >]

[B08B6D39FM]

Carol Wright Gifts Men's Comfy Lounge Pant

14.99 to 17.99

[B075RCSFJ5]

Carol Wright Gifts Men's Fleece Lounge Pants by Cozee Corner

14.99 to 17.99

[B003LUVGVI]

Carol Wright Gifts Women's Flats | Comfortable Flats for Women | Women's Dress Flats

21.99 to 32.99

[B08BJ9VXVY]

Carol Wright Gifts Comfy Slip-On

21.99

. . .

Action: click[b08b6d39fm]

2. Past: Searching for women's shoes with rubber soles in size 7.5 for less than 40.

The interface is:

[Back to Search]

Page 1 (Total results: 50)

[Next >]

[B0751HY5BG]
ASICS Men's Gel-Lyte Gymnastics Shoes
35.34 to 77.99
[B09L5YX71C]
Foggs Japan Slippers, Cozislides Original, CloudFeet Ultra-Soft Slippers, EVA Universal Quick-drying Thickened Non-slip Sandals
100.0
[B09PQXCB3V]
ZSpzx Women's Fashion Sneakers Lace Up Casual Round Toe Arch Support Comfortable Low Top Sneakers Glitter Classic Leather Cushion Foam Comfort Loafers Dress Shoes Ladies Walking Flat Shoes 24.99 to 32.8
. . .
Action: click[b0751hy5bg]
. . .
. . .

**Observation2:**
Past: Searched for black 4x-large polyester cotton loose fit men's shorts with elastic waistband that have long lasting moisture wicking.
The interface is:
[Back to Search]
Page 1 (Total results: 50)
[Next >]
[B09Q5ZHRVM]
Yinimo Mens Gym Shorts Fashionable Loose Beach Drawstring Turkey Print Beach Cool Shorts
19.19
[B09QQP3356]
HAUKLIE Men's Sports Waffle Ribbed Polo Shirts Summer Short Sleeve Cotton Muscle Quarter-Zip Henley T-Shirt Tunics Tops
10.99
[B09ND8P2QR]
InterestPrint Deer and Stars Men's Loungewear Pajama Sets - Long Sleeve Tee and Jogger Pant
43.59
**Action2:** click[B09Q5ZHRVM]
**Prompt3:**
1.Past: Searched for black ankle strap flats in size 9 wide and clicked on b09r2dhbd3.
The interface is:
[Back to Search]
[< Prev]
color [aa04-black] [aa04-brown] [aa04-khaki] [aa03-beige] [aa03-black] [aa02-yellow] [aa-01 beige] [aa-01 brown]
size [5.5] [6] [6.5] [6.5-7] [7] [7.5] [8] [8.5] [9] [9.5-10] [10] [10.5] [11] [11.5]
Women Flats Sandals for Summer,Fashion Casual Flat Shoes Solid Color Denim Zipper Sandals Buckle Strap Open Toe Flip Flops
Price: 7.07 to 8.23
Rating: N.A.
[Description]
[Features]
[Reviews]
[Buy Now]
Action: click[aa03-black]
2.Past: Searched for black ankle strap flats in size 9 wide, clicked on b09r2dhbd3, and then clicked on aa03-black.

The interface is:
[Back to Search]
[< Prev]
color [aa04-black] [aa04-brown] [aa04-khaki] [aa03-beige] [aa03-black] [aa02-yellow] [aa-01 beige]
[aa-01 brown]
size [5.5] [6] [6.5] [6.5-7] [7] [7.5] [8] [8.5] [9] [9.5-10] [10] [10.5] [11] [11.5]
Women Flats Sandals for Summer,Fashion Casual Flat Shoes Solid Color Denim Zipper Sandals
Buckle Strap Open Toe Flip Flops
Price: 7.07 to 8.23
Rating: N.A.
[Description]
[Features]
[Reviews]
[Buy Now]
Action: click[9]
. . .

. . .

**Observation3:**
Past: Searched for and clicked on black 4x-large polyester cotton men's shorts with elastic waistband
that have long lasting moisture wicking and a loose fit.
The interface is:
[Back to Search]
[< Prev]
color [black] [blue] [red]
size [small] [medium] [large] [x-large] [xx-large] [3x-large] [4x-large] [5x-large]
Yinimo Mens Gym Shorts Fashionable Loose Beach Drawstring Turkey Print Beach Cool Shorts
Price: 19.19
Rating: N.A.
[Description]
[Features]
[Reviews]
[Buy Now]
**Action3:** click[black]
**Prompt4:**
. . .

2.Past: Searched for black ankle strap flats in size 9 wide, clicked on b09r2dhbd3, and then clicked on
aa03-black.
The interface is:
[Back to Search]
[< Prev]
color [aa04-black] [aa04-brown] [aa04-khaki] [aa03-beige] [aa03-black] [aa02-yellow] [aa-01 beige]
[aa-01 brown]
size [5.5] [6] [6.5] [6.5-7] [7] [7.5] [8] [8.5] [9] [9.5-10] [10] [10.5] [11] [11.5]
Women Flats Sandals for Summer,Fashion Casual Flat Shoes Solid Color Denim Zipper Sandals
Buckle Strap Open Toe Flip Flops
Price: 7.07 to 8.23
Rating: N.A.
[Description]
[Features]
[Reviews]
[Buy Now]

Action: click[9]

3.Past: Searched for black ankle strap flats in size 9 wide, clicked on b09r2dhbd3, then clicked on aa03-black and finally clicked on size 9.

The interface is:

[Back to Search]

[< Prev]

color [aa04-black] [aa04-brown] [aa04-khaki] [aa03-beige] [aa03-black] [aa02-yellow] [aa-01 beige] [aa-01 brown]

size [5.5] [6] [6.5] [6.5-7] [7] [7.5] [8] [8.5] [9] [9.5-10] [10] [10.5] [11] [11.5]

Women Flats Sandals for Summer,Fashion Casual Flat Shoes Solid Color Denim Zipper Sandals Buckle Strap Open Toe Flip Flops

Price: 7.07 to 8.23

Rating: N.A.

[Description]

[Features]

[Reviews]

[Buy Now]

Action: click[buy now]

. . .

**Observation4:**

Past: Searched for and clicked on black, long-lasting, moisture-wicking, loose-fit men's shorts with elastic waistband in 4X-large size made of polyester cotton.

The interface is:

[Back to Search]

[< Prev]

color [black] [blue] [red]

size [small] [medium] [large] [x-large] [xx-large] [3x-large] [4x-large] [5x-large]

Yinimo Mens Gym Shorts Fashionable Loose Beach Drawstring Turkey Print Beach Cool Shorts

Price: 19.19

Rating: N.A.

[Description]

[Features]

[Reviews]

[Buy Now]

You have clicked black.

**Action4:** click[4x-large]

**Prompt5:**

. . .

1.Past: Past: Searched for black ankle strap flats in size 9 wide, clicked on b09r2dhbd3, then clicked on aa03-black and finally clicked on size 9.

The interface is:

[Back to Search]

[< Prev]

color [aa04-black] [aa04-brown] [aa04-khaki] [aa03-beige] [aa03-black] [aa02-yellow] [aa-01 beige] [aa-01 brown]

size [5.5] [6] [6.5] [6.5-7] [7] [7.5] [8] [8.5] [9] [9.5-10] [10] [10.5] [11] [11.5]

Women Flats Sandals for Summer,Fashion Casual Flat Shoes Solid Color Denim Zipper Sandals Buckle Strap Open Toe Flip Flops

Price: 7.07 to 8.23

Rating: N.A.

[Description]
[Features]
[Reviews]
[Buy Now]
Action: click[buy now]
2.Past: Searched for 5XL Carol Wright men's lounge pants in charcoal color, clicked on item code B08B6D39FM, selected size 5X-large, and chose the color charcoal.
The interface is:
[Back to Search]
[< Prev]
size [small] [medium] [large] [x-large] [xx-large] [3x-large] [4x-large] [5x-large ]
color [black] [charcoal] [gray] [heather blue] [navy] [plaid] [olive]
Carol Wright Gifts Men's Comfy Lounge Pant
Price: 14.99 to 17.99
Rating: N.A.
[Description]
[Features]
[Reviews]
[Buy Now]
Action: click[buy now]
. . .
. . .
**Observation5:**
Past: Searched for and clicked on black 4x-large polyester cotton moisture wicking loose fit men's shorts with elastic waistband (ASIN: B09Q5ZHRVM).
The interface is:
[Back to Search]
[< Prev]
color [black] [blue] [red]
size [small] [medium] [large] [x-large] [xx-large] [3x-large] [4x-large] [5x-large]
Yinimo Mens Gym Shorts Fashionable Loose Beach Drawstring Turkey Print Beach Cool Shorts
Price: 19.19
Rating: N.A.
[Description]
[Features]
[Reviews]
[Buy Now]
You have clicked black.
You have clicked 4x-large.
**Action5:** click[Buy Now]

Table 5: An Example decision process in the WebShop.

**Goals:** cool some apple and put it in diningtable.

You are in the middle of a room. Looking quickly around you, you see a cabinet 1, a cabinet 10, a cabinet 11, a cabinet 2, a cabinet 3, a cabinet 4, a cabinet 5, a cabinet 6, a cabinet 7, a cabinet 8, a cabinet 9, a coffeemachine 1, a countertop 1, a countertop 2, a diningtable 1, a drawer 1, a drawer 2, a drawer 3, a fridge 1, a garbagecan 1, a microwave 1, a sinkbasin 1, a stoveburner 1, a stoveburner 2, a stoveburner 3, a stoveburner 4, and a toaster 1.

**Prompt1:**

1. Observation: You are in the middle of a room. Looking quickly around you, you see nothing.
Evaluation:Past: Subgoal 1: - Incomplete Subgoal 2: Cool - Incomplete Subgoal 3: - Incomplete
Summary: Unknown. More information is needed to provide a summary of the past process.
Action: go to sinkbasin 1
2. Observation: You arrive at loc 12. On the sinkbasin 1, you see a apple 2, a mug 2, and a potato 1.
Evaluation: Past: Subgoal 1: Take apples from somewhere - Incomplete Subgoal 2: Cool apples - Incomplete Subgoal 3: Put apples into dining table - Incomplete
Summary: Looked and went to sink basin 1.
Action: take apple 2 from sinkbasin 1
3. Observation: You arrive at loc 8. On the sinkbasin 1, you see a egg 1, a mug 1, and a potato 2.
Evaluation:Past: Subgoal 1: Take apple from somewhere - Incomplete Subgoal 2: Cool apple - Incomplete Subgoal 3: Put apple into countertop - Incomplete
Summary: Looked and went to sink basin 1.
Action: go to microwave 1
4. Observation: You arrive at loc 10. The microwave 1 is closed.
Evaluation:Past: Subgoal 1: Take apple from somewhere - Incomplete Subgoal 2: Cool apple - Incomplete Subgoal 3: Put apple into countertop - Incomplete
Summary: Looked and then went to sink basin 1 and microwave 1.
Action: open microwave 1
5. Observation: You close the microwave 1.
Evaluation:Past: subgoal 1: take apple from somewhere - incomplete subgoal 2: cool apple - incomplete subgoal 3: put apple into countertop - incomplete
Summary: Looked and went to sink basin 1, then went to microwave 1, opened and closed it.
Action: go to garbagecan 1
. . .
. . .

**Observation1:**

Observation: You are in the middle of a room. Looking quickly around you, you see nothing.
Evaluation: Subgoal 1: take apple from somewhere -Incomplete Subgoal 2: cool apple -Incomplete Subgoal 3: put apple into countertop -Incomplete
Summary: Unknown. More information is needed to provide a summary of the past process.
**Action1:** go to garbagecan 1

**Prompt2:**

1. Observation: You are in the middle of a room. Looking quickly around you, you see nothing.
Evaluation:Past: Subgoal 1: - Incomplete Subgoal 2: Cool - Incomplete Subgoal 3: - Incomplete
Summary: Unknown. More information is needed to provide a summary of the past process.
Action: go to sinkbasin 1
2. Observation: You arrive at loc 12. On the sinkbasin 1, you see a apple 2, a mug 2, and a potato 1.
Evaluation: Past: Subgoal 1: Take apples from somewhere - Incomplete Subgoal 2: Cool apples - Incomplete Subgoal 3: Put apples into dining table - Incomplete
Summary: Looked and went to sink basin 1.
Action: take apple 2 from sinkbasin 1

3. Observation: You arrive at loc 8. On the sinkbasin 1, you see a egg 1, a mug 1, and a potato 2.

Evaluation:Past: Subgoal 1: Take apple from somewhere - Incomplete Subgoal 2: Cool apple - Incomplete Subgoal 3: Put apple into countertop - Incomplete

Summary: Looked and went to sink basin 1.

Action: go to microwave 1

4. Observation: You arrive at loc 10. The microwave 1 is closed.

Evaluation:Past: Subgoal 1: Take apple from somewhere - Incomplete Subgoal 2: Cool apple - Incomplete Subgoal 3: Put apple into countertop - Incomplete

Summary: Looked and then went to sink basin 1 and microwave 1.

Action: open microwave 1

5. Observation: You close the microwave 1.

Evaluation:Past: subgoal 1: take apple from somewhere - incomplete subgoal 2: cool apple - incomplete subgoal 3: put apple into countertop - incomplete

Summary: Looked and went to sink basin 1, then went to microwave 1, opened and closed it.

Action: go to garbagecan 1

. . .

. . .

**Observation2:**

Observation: On the garbagecan 1, you see a apple 1, and a tomato 1.

Evaluation: Subgoal 1: take apple from somewhere -Incomplete Subgoal 2: cool apple -Incomplete Subgoal 3: put apple into countertop -Incomplete

Summary: Looked and then went to garbagecan 1.

**Action2:** take apple 1 from garbagecan 1

**Prompt3:**

1. Observation: You pick up the apple 2 from the sinkbasin 1.

Evaluation:Past: Subgoal 1: Take apple from somewhere - Complete Subgoal 2: Cool apple - Incomplete Subgoal 3: Put apple into dining table - Incomplete

Summary: Took apple 2 from sinkbasin 1.

Action: go to fridge 1

2. Observation: You arrive at loc 16. The fridge 1 is closed.

Evaluation:Past: Subgoal 1: Take apple from somewhere - Complete Subgoal 2: Cool apple - Incomplete Subgoal 3: Put apple into dining table - Incomplete

Summary: Looked, went to sinkbasin 1, took apple 2 from sinkbasin 1, and went to fridge 1.

Action: cool apple 2 with fridge 1

3. Observation: You pick up the apple 1 from the garbagecan 1.

Evaluation:Past: Subgoal 1: Take apple from somewhere - Complete Subgoal 2: Cool apple - Incomplete Subgoal 3: Put apple into countertop - Incomplete

Summary: Looked around, went to sinkbasin 1, went to microwave 1, opened and closed microwave 1, went to garbagecan 1, took apple 1 from garbagecan 1.

Action: go to fridge 1

4. Observation: You arrive at loc 1. The fridge 1 is closed.

Evaluation:Past: Subgoal 1: Take apple from somewhere - Complete Subgoal 2: Cool apple - Incomplete Subgoal 3: Put apple into countertop - Incomplete

Summary: Looked around, went to sinkbasin 1, went to microwave 1, opened and closed microwave 1, went to garbagecan 1, took apple 1 from garbagecan 1, and went to fridge 1.

Action: cool apple 1 with fridge 1

. . .

. . .

**Observation3:**
Observation: You pick up the apple 1 from the garbagecan 1.
Evaluation: Subgoal 1: take apple from somewhere -complete Subgoal 2: cool apple -Incomplete Subgoal 3: put apple into countertop -Incomplete
Summary: Looked for apple, found it in garbage can 1.
**Action3:** go to fridge 1
**Prompt4:**
1. Observation: You pick up the apple 2 from the sinkbasin 1.
Evaluation:Past: Subgoal 1: Take apple from somewhere - Complete Subgoal 2: Cool apple - Incomplete Subgoal 3: Put apple into dining table - Incomplete
Summary: Took apple 2 from sinkbasin 1.
Action: go to fridge 1
2. Observation: You arrive at loc 16. The fridge 1 is closed.
Evaluation:Past: Subgoal 1: Take apple from somewhere - Complete Subgoal 2: Cool apple - Incomplete Subgoal 3: Put apple into dining table - Incomplete
Summary: Looked, went to sinkbasin 1, took apple 2 from sinkbasin 1, and went to fridge 1.
Action: cool apple 2 with fridge 1
3. Observation: You pick up the apple 1 from the garbagecan 1.
Evaluation:Past: Subgoal 1: Take apple from somewhere - Complete Subgoal 2: Cool apple - Incomplete Subgoal 3: Put apple into countertop - Incomplete
Summary: Looked around, went to sinkbasin 1, went to microwave 1, opened and closed microwave 1, went to garbagecan 1, took apple 1 from garbagecan 1.
Action: go to fridge 1
4. Observation: You arrive at loc 1. The fridge 1 is closed.
Evaluation:Past: Subgoal 1: Take apple from somewhere - Complete Subgoal 2: Cool apple - Incomplete Subgoal 3: Put apple into countertop - Incomplete
Summary: Looked around, went to sinkbasin 1, went to microwave 1, opened and closed microwave 1, went to garbagecan 1, took apple 1 from garbagecan 1, and went to fridge 1.
Action: cool apple 1 with fridge 1
. . .
. . .
**Observation4:**
Observation: The fridge 1 is closed.
Evaluation: Subgoal 1: take apple from somewhere -complete Subgoal 2: cool apple -Incomplete Subgoal 3: put apple into countertop -Incomplete
Summary: Looked for apple, found it in garbage can 1 and took it.
**Action4:** cool apple 1 with fridge 1
**Prompt5:**
1. Observation: You cool the apple 2 using the fridge 1.
Evaluation:Past: subgoal 1: take apple 2 from somewhere - complete subgoal 2: cool apple 2 - complete subgoal 3: put apple 2 into diningtable - incomplete
Summary: Looked for apple 2, took it from sinkbasin 1, and cooled it with fridge 1.
Action: go to diningtable 1
2. Observation: You arrive at loc 15. On the diningtable 1, you see a apple 3, a apple 4, a bowl 1, a bowl 2, a butterknife 1, a egg 3, a fork 2, a fork 3, a knife 2, a knife 3, a lettuce 1, a lettuce 2, a lettuce 3, a spatula 3, and a spoon 2.

Evaluation:Past: Subgoal 1: Take apple 2 from somewhere - Complete Subgoal 2: Cool apple 2 - Complete Subgoal 3: Put apple 2 into diningtable 1 - Complete

Summary: Looked for apple 2, took it from sinkbasin 1, cooled it in fridge 1, and placed it on diningtable 1.

Action: put apple 2 in/on diningtable 1

3. Observation: You cool the apple 1 using the fridge 1.

Evaluation:Past: Subgoal 1: Take apple from somewhere - Complete Subgoal 2: Cool apple - Complete Subgoal 3: Put apple into countertop - Incomplete

Summary: Looked for an apple, went to sinkbasin 1, went to microwave 1, opened and closed microwave 1, went to garbagecan 1, took apple 1 from garbagecan 1, went to fridge 1, cooled apple 1 with fridge 1.

Action: go to countertop 1

4. Observation: You arrive at loc 15. On the countertop 1, you see a bowl 1, a bread 1, a butterknife 1, a cup 1, a dishsponge 1, a fork 2, a knife 2, a ladle 2, a pan 1, a potato 1, a soapbottle 1, a spatula 1, a spatula 2, a spatula 3, and a spoon 1.

Evaluation:Past: Subgoal 1: Take apple from somewhere - Complete Subgoal 2: Cool apple - Complete Subgoal 3: Put apple into countertop - Complete

Summary: Looked for an apple, found one in the garbage, cooled it in the fridge, and placed it on the countertop.

Action: put apple 1 in/on countertop 1

. . .

. . .

**Observation5:**

Observation: You cool the apple 1 using the fridge 1.

Evaluation: Subgoal 1: take apple from somewhere -complete Subgoal 2: cool apple -complete Subgoal 3: put apple into countertop -Incomplete

Summary: Looked for apple 1, took it from garbage can 1, and cooled it with fridge 1.

**Action5:** go to countertop 1

**Prompt6:**

1. Observation: You cool the apple 2 using the fridge 1.

Evaluation:Past: subgoal 1: take apple 2 from somewhere - complete subgoal 2: cool apple 2 - complete subgoal 3: put apple 2 into diningtable - incomplete

Summary: Looked for apple 2, took it from sinkbasin 1, and cooled it with fridge 1.

Action: go to diningtable 1

2. Observation: You arrive at loc 15. On the diningtable 1, you see a apple 3, a apple 4, a bowl 1, a bowl 2, a butterknife 1, a egg 3, a fork 2, a fork 3, a knife 2, a knife 3, a lettuce 1, a lettuce 2, a lettuce 3, a spatula 3, and a spoon 2.

Evaluation:Past: Subgoal 1: Take apple 2 from somewhere - Complete Subgoal 2: Cool apple 2 - Complete Subgoal 3: Put apple 2 into diningtable 1 - Complete

Summary: Looked for apple 2, took it from sinkbasin 1, cooled it in fridge 1, and placed it on diningtable 1.

Action: put apple 2 in/on diningtable 1

3. Observation: You cool the apple 1 using the fridge 1.

Evaluation:Past: Subgoal 1: Take apple from somewhere - Complete Subgoal 2: Cool apple - Complete Subgoal 3: Put apple into countertop - Incomplete

Summary: Looked for an apple, went to sinkbasin 1, went to microwave 1, opened and closed microwave 1, went to garbagecan 1, took apple 1 from garbagecan 1, went to fridge 1, cooled apple 1 with fridge 1.

Action: go to countertop 1

4. Observation: You arrive at loc 15. On the countertop 1, you see a bowl 1, a bread 1, a butterknife 1, a cup 1, a dishsponge 1, a fork 2, a knife 2, a ladle 2, a pan 1, a potato 1, a soapbottle 1, a spatula 1, a spatula 2, a spatula 3, and a spoon 1.
Evaluation:Past: Subgoal 1: Take apple from somewhere - Complete Subgoal 2: Cool apple - Complete Subgoal 3: Put apple into countertop - Complete
Summary: Looked for an apple, found one in the garbage, cooled it in the fridge, and placed it on the countertop.
Action: put apple 1 in/on countertop 1
· · ·
· · ·
**Observation6:**
Observation: You arrive at loc 15. On the countertop 1, you see a bowl 1, a bread 1, a butterknife 1, a cup 1, a dishsponge 1, a fork 2, a knife 2, a ladle 2, a pan 1, a potato 1, a soapbottle 1, a spatula 1, a spatula 2, a spatula 3, and a spoon 1.
Evaluation: Subgoal 1: take apple from somewhere -complete Subgoal 2: cool apple -complete Subgoal 3: put apple into countertop -complete
Summary: Looked for apple 1, took it from garbage can 1, cooled it in fridge 1, and placed it on countertop 1.
**Action6:** put apple 1 in/on diningtable 1

Table 6: An Example decision process in the ALFWorld.