# OpenReview forum: "LDM$^2$: A Large Decision Model Imitating Human Cognition with Dynamic Memory Enhancement"
_EMNLP/2023/Conference — EMNLP 2023 Findings_

### Official Review · Reviewer_c7Lp · 2023-07-27

**Soundness:** 3

**Excitement:**

3: Ambivalent: It has merits (e.g., it reports state-of-the-art results, the idea is nice), but there are key weaknesses (e.g., it describes incremental work), and it can significantly benefit from another round of revision. However, I won't object to accepting it if my co-reviewers champion it.

**Paper Topic And Main Contributions:**

This paper proposes a dynamic memory mechanism to improve LLMs' decision-making with prompts. The proposed method consists of two stages: the memory formation stage, which stores human behaviors or expert trajectories as state-action tuples, and the memory refinement stage, which employs tree exploration to generate valuable decision processes and dynamically update the memory. The effectiveness of this approach is demonstrated in Webshop and ALFworld, where it is compared against standard prompting methods, ReAct and BUTLER.

**Questions For The Authors:**

1. What are the advantages of the proposed method compared to other memory-based frameworks for reinforce language agents, such as Reflexion [1]? Besides, it is necessary to add baselines about other state-of-the-art memory-based frameworks to demonstrate this.
1. I was wondering will later valuable state-action tuples replace the prior ones, or is there a way to limit the size of the memory from being explosive?
1. It seems that the chosen action depends on the history information of state-action tuples from the initial timestep. Is there a possibility of reducing the horizon of prompts for LLMs while maintaining the same level of performance to save the in-context learning budget?

[1] Shinn, Noah, Beck Labash, and Ashwin Gopinath. "Reflexion: an autonomous agent with dynamic memory and self-reflection." arXiv preprint arXiv:2303.11366 (2023).

**Reasons To Accept:**

In terms of novelty, this paper investigates an interesting problem, capitalizing on the remarkable summarization abilities of LLMs and dynamic prompting mechanisms to boost their decision-making performance. Unlike fixed or hand-crafted prompts, the proposed dynamic memory offers a natural and self-updating prompting approach, encouraging LLMs to improve their decisions based on environmental feedback. The experimental results demonstrate the effectiveness of the proposed method.

In terms of clarity, this paper is largely clear and concise. The problem setting and the ﻿two-stage memory mechanism are easy to follow. Moreover, the figures, in general, are clean and easy to understand.

**Reasons To Reject:**

As mentioned in the limitation section, I am also concerned about the applicability of the proposed method in other decision-making problems when the human trajectories or expert examples are insufficient. Besides, when the action space is larger, the tree exploration process could be time-consuming. Maybe some suggestions about how to integrate other tree search methods could help alleviate this issue.

**Reproducibility:**

5: Could easily reproduce the results.

**Reviewer Confidence:**

4: Quite sure. I tried to check the important points carefully. It's unlikely, though conceivable, that I missed something that should affect my ratings.

**Typos Grammar Style And Presentation Improvements:**

There are some grammatical errors and confusing sentences in this paper mainly because of inaccurate expressions. It is necessary for the authors to check the whole paper and correct them. Just some examples:
Line 37: have->has
Line 112: spaces->space
Line 166: employed and Line 175: learned. Tenses in related work need to be unified.
Line 200: it forms->which forms
Line 220: the the-> the
The paragraph from Line 368 to Line 373 is confusing.
Line 572: e->we?

---

> ### Author Rebuttal · Authors · 2023-08-27
>
> Thanks for your suggestions.
>
> Actually, when the human data are insufficient, the performance will be affected, but our method could still work when there are only few trajectories (ten or less). In such cases, the existing methods just take these few examples as prompt which is fixed at each timestep, but our method splits these examples into state-action tuples based on the observation then constructs dynamic prompt in each timestep.
>
> Thanks again for your valuable suggestion on the tree exploration. We consider adding an evaluation process in the tree exploration process which lets the LLMs agent evaluate whether the current exploration direction is worth continuing. If the current exploration is valueless, the agent will prune the subtree from there.
>
> Q1: First, the stage is different. The memory of Reflexion and other existing methods mainly focus on recording the experience and reflection on the online exploration phase. But our method leverages human data to form an offline initial memory to make the LLM agent familiar with the task in advance and then refines the memory according to the online exploration. The initial memory is a crucial part of our method, it leverages human data to make the agent construct an initial world model.
> Second, the information in the memory is different. The Reflexion and other existing methods record the past experience or reflections as the memory which are non-uniform or nonstandard, but our memory consists of standard state-action tuples (please refer to Section 3.2). which is easier to store, refine, and manipulate. Meanwhile, we cluster the standard data in the memory to form the index of data, which is beneficial for standard data operation (such as add, revise, and delete). Existing methods just directly add the nonstandard text into the memory and the resulting memory is hard to refine and retrieve.
> We have cited the Reflexion in the related work. Reflexion is not a competitive approach, but it can be integrated into our method to further improve the performance. Reflexion uses self-reflect from the completed trajectory and repeats the same task again based on the reflection, which could be adapted to our method that leverage the self-reflection to make multiple decision trajectories on the same instance. We will conduct such experiment and report its results.
>
> Q2: In the current setting, we directly add the valuable state-action tuples into the corresponding sub-memory. We do not replace any data because they are different blocks of trajectories from different instances. In practice, if the initial memory is small, the exploration stage will produce a lot of valuable state-action tuples. However, when the memory size reaches a certain scale, the agent may accomplish most instances of certain task, and it can hardly find new valuable state-action tuples. If the human data is sufficient and the initial memory is large, the new valuable state-action tuples will be few and the memory size tends to be stable.
>
> Q3: The state-action tuples consist of goals, summarized history information, and the current observation. These three parts are equally important, the goal tells the agent the decision-making direction, the current observation teaches the agent how to act with similar observation, and the history information tells the agent what have done to get the current state. Different paths may reach the same state, but they are not the same state-action tuples. The history information could assist the agent finding the similar experience and avoiding wrong actions appeared in the past.

---

### Official Review · Reviewer_gqCr · 2023-08-01

**Soundness:** 2

**Excitement:**

2: Mediocre: This paper makes marginal contributions (vs non-contemporaneous work), so I would rather not see it in the conference.

**Paper Topic And Main Contributions:**

Designing appropriate prompts for LLMs is difficult because of the lack of generalization in a complex environment. Merely imitating prompting examples, the traditional prompting method cannot adapt to situations where the environment content is distinctive from the examples.

This paper proposes establishing and maintaining a memory-based trajectory library that records state-action tuples and other information. The main idea is composed of two phrases: memory formation and memory refinement. In the memory formation phase, human trajectories are summarized into state-action tuples by LLMs and clustered with goal labels and observation labels. In the refinement phase, LDM2 explores the tree structure to retrieve the relevant information, and later LDM2 is finetuned by adding high-reward state-action tuples.

**Reasons To Accept:**

- The idea of LDM2 is quite reasonable. Intuitively, increasing the memory of the trajectories (candidate prompt components) helps get optimal prompts. However, larger is not always better. The main contribution of LDM2 lies in constructing a clustered memory and reward-based refinement, which dynamically updates the memory content towards a higher-reward state.

- LDM2 constructs a tree exploration method to get all possible excellent state-action tuples updating the memory. In some cases, choosing which action are ambiguous due to the similarity of observation and historical information, so choosing multi-action is practical.

**Reasons To Reject:**

- Exploring possible trajectories with tree structures helps find high-reward trajectories. However, if there are numerous actions to explore as well as in subnodes, it will be computation-consuming since the complexity is exponential.

- In the Memory Formation phase, state-action tuples are clustered by LLM. But we all know that LLM is not fully accurate and fails to get the right answer. If LLM fails in the clustering phase, it may affect the continual process. So would it be better to substitute LLM with the traditional clustering method?

- The experiments are all done with GPT-3.5, which is relatively weak. It is possible that with GPT4 and longer context windows, the proposed framework does not provide any noticeable improvement.

**Reproducibility:**

3: Could reproduce the results with some difficulty. The settings of parameters are underspecified or subjectively determined; the training/evaluation data are not widely available.

**Reviewer Confidence:**

3: Pretty sure, but there's a chance I missed something. Although I have a good feel for this area in general, I did not carefully check the paper's details, e.g., the math, experimental design, or novelty.

---

> ### Author Rebuttal · Authors · 2023-08-27
>
> Thanks for your thoughtful comments and suggestions.
>
> 1.If there are numerous actions in each subnode, the exploration paths will indeed grow exponentially.  In our experiments, we find that the action distribution of retrieved state-action tuples in each node is highly concentrative and we instruct the LLM agent to generate the most occurred actions which are relatively few (only one in most cases). What’s more, nodes with several forks are mainly located in the upper layers of the exploration tree, and nodes in the lower layers are basically not needed to split. Hence, the exploration tree will not grow exponentially and the cost is considerable. Meanwhile, we consider adding an evaluation process in the tree exploration process which lets the LLM agent evaluate whether the current exploration direction is worth continuing. If the current exploration is valueless, the agent will prune the subtree from this node.
>
> 2.Comparing with traditional clustering methods, our method has the following advantages. First, LLMs work with text-based input and generate text-based output, instead of numerical data representations, which is more flexible. Second, LLMs have the potential to capture complex semantic relationships between data points based on textual descriptions; LLMs consider not only quantitative features, but also textual context. Hence, using LLMs for clustering is more effective in text-rich domains where their language understanding capability can be harnessed effectively. What’s more, the LLM agent leverage its own in-parametric knowledge and cognitive ability to cluster the tuples, the clustering result is also beneficial to the LLM agent when faced with retrieving and understanding tasks in the future, although the result is not completely correct in the point of view of human being. The LLM agent is completely independent, it can act as versatile tools to accomplish sub-tasks and complete the whole decision-making process without resorting to other external tools.
>
> 3.The GPT-4 API was released after the deadline of the submission phase, so we only used  GPT-3.5 in the experiment. We will provide the GPT-4 results in the future.

---

### Official Review · Reviewer_X6GX · 2023-08-03

**Soundness:** 3

**Excitement:**

3: Ambivalent: It has merits (e.g., it reports state-of-the-art results, the idea is nice), but there are key weaknesses (e.g., it describes incremental work), and it can significantly benefit from another round of revision. However, I won't object to accepting it if my co-reviewers champion it.

**Missing References:**

the procedure in L338-342 resembles Zhao et al. 2023 (https://aclanthology.org/2023.acl-long.320/).


**Paper Topic And Main Contributions:**

This work proposes LDM^2, a large decision model with memory. It uses state-action tuples derived from human behaviors to form a dynamic memory storage. Then, it leverages relevant memory to guide the LLM in its decision process. It searches all potential decision processes with tree-exploration and evaluates them using environment rewards. Then, the best decision process is selected and added to dynamic memory. The main innovation is the exploration-evaluation-adding circle, which mimics reinforcement learning.


**Questions For The Authors:**

1. Refer to Con(1), could you provide a more detailed cost analysis and comparison to existing methods?
2. Refer to Con(2), did you have to collect human data? If so, you could give more details on how they are collected, who they are collected from, etc.

3. The human behaviors are not exactly “training data” as no real training happens during the pipeline? Maybe replace this term to avoid confusion?
4. I may have missed this, but I don’t see how the reward $r$ is calculated from the methodology section?
5. The memory formation stage requires rewriting the human behaviors into a summary of state-action tuples with the GPT model. During this process, would there be any information loss? Are there any mistakes made during rewriting? I’d love to see an analysis on this aspect as well.


**Reasons To Accept:**

1. The structure is innovative. The authors attempt to mimic the RL algorithm by designing different components that utilizes LLMs, such as dynamic memory, decoding (tree exploration in this case).
2. The results show good improvement from the baselines.


**Reasons To Reject:**

1. This algorithm may incur high costs. Firstly, the LLM needs to encode all training data (human behaviors) into state-action tuples. Secondly, due to the context length constraint, a decision process needs to be derived from each batch memory (M_b) in order to select the optimal process. Thirdly, the tree structure search may incur higher costs than normal decoding strategies as many paths need to be maintained simultaneously. Overall, even though the results are improved, it may not be worth it if the costs have increased too much.
2. To provide better data, i.e. human behaviors, human responses need to be collected, which may again be very costly.


**Reproducibility:**

3: Could reproduce the results with some difficulty. The settings of parameters are underspecified or subjectively determined; the training/evaluation data are not widely available.

**Reviewer Confidence:**

4: Quite sure. I tried to check the important points carefully. It's unlikely, though conceivable, that I missed something that should affect my ratings.

---

> ### Author Rebuttal · Authors · 2023-08-27
>
> Thanks for your thoughtful comments and suggestions.
>
> 1.The number of training data of each task is not more than 200, thus it is not very difficult for LLMs to generate state-action tuples. The cost is medium and worth, because the agent can automatically learn from the given training data for different tasks instead of carefully designing task-specific prompts by hand. This process could also assist the agent to form a task-specific initial memory.
>
> 2.Due to the context length constraint, we have to split the whole training data into several batches. Moreover, since the training data is not evenly distributed, each batch should be considered. Besides, the computational cost of this stage also depends on the number of training data, but the overall cost is not much.
>
> 3.If there are many diverging actions in each node, the exploration paths will indeed grow exponentially. In our experiments, we find that the action distribution of retrieved state-action tuples in each node is highly concentrative and we instruct the LLM agent to generate the most occurred actions which are relatively few (only one in most cases). What’s more, nodes with several forks are mainly located in the upper layers of the exploration tree, and nodes in the lower layers are basically not needed to split. Hence, the exploration tree will not grow exponentially and the cost is considerable. Meanwhile, we consider adding an evaluation process in the tree exploration process which lets the LLM agent evaluate whether the current exploration direction is worth continuing. If the current exploration is valueless, the agent will prune the subtree from this node.
> 4.	The number of human training data is not more than 200 which is not difficult to collect. In fact, there are plenty of available data collected in many scenarios. One innovation of our method is that we use these human data to automatically construct prompts.
>
> Q1: We can compare the time cost and computation cost of our method with existing methods. At each timestep, existing methods only need one inference to generate action but our method need to instruct LLMs to retrieve relevant memory and then generate action. In average, the number of LLM API calls of our method is four times than that of existing methods. However, the prompt length of each API call of our method is shorter because  only the current state needs to be considered, but existing methods must record the whole past experience which will increase as time goes by. The amount of inference tokens of our method and existing methods are roughly equal which is shown by the billing provided by OpenAI. All in all, the computation cost of our method is comparable with other methods, but our method bears a higher time cost.
>
> Q2: In our experiment, we directly use the human data collected by Yao et al. (2022a) for WebShop and Shridhar et al. (2020) for ALFworld.
>
> Q3: The reason why we refer the human behaviors as “training data” is that we feed these data to the LLM agents, and the agents learn from these data to form the memory that can be used to accomplish the task. The quality and number of the samples will affect the performance of learning. This process mimics the training process as the context of the memory is varying, though there is no parameter updating.
>
> Q4: We directly use the WebShop reward defined in Yao et al. (2022a) and the ALFworld reward defined in Shridhar et al. (2020).
>
> Q5: The state-action tuples consist of task goal g, history information ht, and current observation ot. The content length of these three should be roughly equal, otherwise the LLM agent may pay too much attention to one part and make unreasonable actions. We summarize the history information into short text with the same order of magnitude in length as the current observation, thus the agent pays equal attention to all parts in the tuples and considers them comprehensively. The summary process indeed loss some information, but the details in the history is not crucial and the LLMs agent can use the details in the current observation to help make decisions. The summarized history information assists the agent finding the similar experience and avoiding wrong actions appeared in the past.

---

### Meta-Review · Area_Chair_mpKD · 2023-09-19

**Recommendation:** 3

**Metareview:**

In complex test environments where the environment content is distinctive from the example environments, traditional prompting methods suffer from lack of generalization.
This paper proposes establishing and maintaining a memory-based trajectory library that records state-action tuples and other information. The main idea is composed of two phrases: memory formation and memory refinement. In the memory formation phase, human trajectories are summarized into state-action tuples by LLMs and clustered with goal labels and observation labels. In the refinement phase, LDM2 explores the tree structure to retrieve the relevant information, and later LDM2 is finetuned by adding high-reward state-action tuples.

Pros:

Intuitive and interesting idea of using memory for dynamic prompting to boost decision making ability of LLMs

Unlike fixed or hand-crafted prompts, the proposed dynamic memory offers a natural and self-updating prompting approach, encouraging LLMs to improve their decisions based on environmental feedback.

The experimental results demonstrate the effectiveness of the proposed method, with a good improvement over baselines

Cons:

In realistic applications of this, the computational cost can be quite high (as pointed out by most of the reviewers)


Again for many realistic applications, getting the expert human behavior trajectories might be quite difficult or  very expensive. Referring to the authors response on this (link), it is important to understand how effective the model would be if only a few example trajectories are provided for a test environment. Some more experimental analysis and ablations are needed to establish this.


What are the advantages of the proposed method compared to other memory-based frameworks for reinforce language agents, such as Reflexion [1]? Besides, it is necessary to add baselines about other state-of-the-art memory-based frameworks to demonstrate this.

---

### Decision · Program_Chairs · 2023-10-07

**Decision:**

Accept-Findings

**Comment:**

In complex test environments where the environment content is distinctive from the example environments, traditional prompting methods suffer from lack of generalization.
This paper proposes establishing and maintaining a memory-based trajectory library that records state-action tuples and other information. The main idea is composed of two phrases: memory formation and memory refinement. In the memory formation phase, human trajectories are summarized into state-action tuples by LLMs and clustered with goal labels and observation labels. In the refinement phase, LDM2 explores the tree structure to retrieve the relevant information, and later LDM2 is finetuned by adding high-reward state-action tuples.

Pros:

Intuitive and interesting idea of using memory for dynamic prompting to boost decision making ability of LLMs

Unlike fixed or hand-crafted prompts, the proposed dynamic memory offers a natural and self-updating prompting approach, encouraging LLMs to improve their decisions based on environmental feedback.

The experimental results demonstrate the effectiveness of the proposed method, with a good improvement over baselines

Cons:

In realistic applications of this, the computational cost can be quite high (as pointed out by most of the reviewers)


Again for many realistic applications, getting the expert human behavior trajectories might be quite difficult or  very expensive. Referring to the authors response on this (link), it is important to understand how effective the model would be if only a few example trajectories are provided for a test environment. Some more experimental analysis and ablations are needed to establish this.


What are the advantages of the proposed method compared to other memory-based frameworks for reinforce language agents, such as Reflexion [1]? Besides, it is necessary to add baselines about other state-of-the-art memory-based frameworks to demonstrate this.